# Dietary Approaches to the Management Of type 2 Diabetes (DIAMOND): protocol for a randomised feasibility trial

Elizabeth Morris,[1,2] Paul Aveyard,[1,2] Pamela Dyson,[2,3] Michaela Noreik,[1,2] Clare Bailey,[4] Robin Fox,[5] Kathy Hoffman,[6] Garry D Tan,[2,3] Susan A Jebb[1,2]

[1]Nuffield Department of Primary Care Health Sciences, University of Oxford, Oxford, UK
[2]NIHR Oxford Biomedical Research Centre, UK
[3]Oxford Centre for Diabetes, Endocrinology and Metabolism (OCDEM), Churchill Hospital, Oxford University Hospitals NHS Foundation Trust and University of Oxford, Oxford, UK
[4]Burnham Health Centre, Slough, UK
[5]Bicester Health Centre, Bicester, UK
[6]Chiltern CCG, Amersham, UK

**Correspondence to**
Dr Elizabeth Morris;
elizabeth.morris@phc.ox.ac.uk

## ABSTRACT

**Introduction** Some clinicians have observed that low-carbohydrate, low-energy diets can improve blood glucose control, with reports of remission from type 2 diabetes in some patients. In clinical trials, support for low-carbohydrate, low-energy diets has been provided by specialist staff and these programmes are unsuitable for widespread deployment in routine primary care. The aim of this trial is to test whether a newly developed behavioural support programme can effectively deliver a low-energy, low-carbohydrate diet in a primary care setting.

**Methods and analysis** This is a feasibility randomised controlled trial (RCT) with embedded qualitative study. Thirty adult patients with type 2 diabetes and body mass index ≥30 kg/m² in 2–4 general practices will be randomised 2:1 intervention or control and followed up over 12 weeks. The intervention diet comprises 8 weeks of a low-carbohydrate food-based diet providing around 800 kcal/day, followed by 4 weeks of weight maintenance. This programme will be delivered by practice nurses, who will also support patients through goal-setting, motivation and self-monitoring across four appointments, and provide a self-help booklet with recipes, shopping lists and other behavioural support. Primary outcome measures of feasibility will be met if CIs do not cross the following proportions: that 60% of intervention group participants attempt the dietary intervention, healthcare professionals conduct the intervention delivery session with at least 60% of essential elements present and 60% of participants attend the final follow-up session. Secondary outcome measures will assess process and qualitative measures, as well as exploratory outcomes including change in haemoglobin A1c and change in weight.

**Ethics and dissemination** This study has been granted ethical approval by the National Research Ethics Service, South Central Oxford B Research Ethics Committee (ref: 18/SC/0071). The study results will inform whether to progress to a full-scale RCT to test the efficacy of offering this programme for patients with type 2 diabetes in primary care.

**Trial registration number** ISRCTN62452621; Pre-results.

## Strengths and limitations of this study

► This study directly addresses a key question regarding the dietary management of patients with type 2 diabetes in routine practice.
► The intervention has been developed by a multidisciplinary team of academic and clinical professionals with patient involvement to create a rigorous yet practical approach suitable for delivery in routine care.
► Quantitative and qualitative analyses will assess the feasibility of the intervention and inform the development of any future full-scale trial.
► This study will not demonstrate the clinical effectiveness of this approach, and a subsequent definitive trial will be required to consider whether this intervention can lead to weight loss, improved glycaemic control and diabetes remission.

## INTRODUCTION

There are an estimated four million people living with diabetes in the UK, with numbers expected to rise to five million by 2025.[1] Globally, almost 15% of all deaths in the 20–79 years age group, are attributable to diabetes.[1] Currently, 10% of the National Health Service (NHS) annual budget is spent on diabetes (around £10 billion). When both direct care and indirect costs are considered, this sum rises to £24 billion, and is predicted to rise further to almost £40 billion by 2035.[1]

Obesity is one of the strongest risk factors for type 2 diabetes,[1] with an estimated 9% increased risk for every 1 kg of weight gained,[2] while even modest weight loss of 5% can improve glycaemic control.[3] For people with established type 2 diabetes, dietary management remains a cornerstone of high-quality care,[4] aiming to lower blood glucose levels and achieve weight loss in those who are overweight.[3] There is strong evidence for the benefit of behavioural interventions in the management of type 2 diabetes; a large, high-quality randomised controlled trial (RCT) of over 5000 adults with type 2 diabetes showed that a 12-month intensive behavioural

intervention (individual and group meetings to support decreased energy intake and increased physical activity) led to a clinically significant weight loss (mean 8.5% of initial weight, compared with 0.6% in the control group), and a significant reduction in mean haemoglobin A1c (HbA1c; −0.6% (0.55mmol/mol), compared with −0.1% (0.1mmol/mol) (control group)) at 1 year.[5] The benefits of the intervention were still evident at 4-year follow-up (weight loss of 4.7% and HbA1c reduction of 0.2% (0.2mmol/mol)).[6] Longitudinal data from the UK Prospective Diabetes Study demonstrates that every 1% (11mmol/mol) reduction in HbA1c is associated with a 21% reduction in risk of diabetes-related mortality, and 37% reduction in risk of microvascular complications.[7]

Weight loss is associated with reduction in HbA1c. Observational analysis of data from RCTs shows that the ORs for a 5%–10% and ≥10% weight loss for a clinically significant 0.5% (0.5mmol/mol) reduction in HbA1c were 3.5 (95% CI 2.8 to 4.4) and 5.4 (4.2 to 7.1), respectively.[8] The recent Diabetes Remission Clinical Trial (DiRECT) trial demonstrated that with a low-energy, total diet replacement programme (850 kcal/day formula diet for 3–5 months), delivered within a primary care setting and designed to achieve 15 kg weight loss, 46% of participants in the intervention group (compared with 4% in the control group, OR 19.7 (7.8 to 49.8)) achieved diabetes remission at 12 months.[9] However, total diet replacement with a formula product is not acceptable to some patients, some of whom may prefer a food-based diet. In addition, food-based dietary strategies may help establish a sustainable diet for the longer term.

There is limited high-quality evidence for the optimal macronutrient intake for patients with diabetes wishing to lose weight, or improve their glycaemic control.[3 10–12] Reductions in glycaemic load, through restriction of total carbohydrate and/or changes in the type of carbohydrate to lower the glycaemic index, may have a beneficial effect, due to the direct effect of carbohydrate consumption on postprandial blood glucose levels,[13] leading to a growing interest in carbohydrate-restricted diets for patients with diabetes. This is reflected in the wealth of recent reviews comparing the effects of low-carbohydrate versus high-carbohydrate diets in patients with diabetes.[11 14–21] Intervention studies have reported that low-carbohydrate diets can reduce HbA1c by mean 0.5% (0.5 mmol/mol), and body weight by mean 4.8 kg, in people with type 2 diabetes.[14 22] Recent data from systematic reviews demonstrate that these low-carbohydrate diets may be safe and effective in the short term, yielding greater reductions in HbA1c at 3 (−0.5%, 95% CI −0.7% to −0.2%) and 6 months (−0.4%, 95% CI −0.6% to −0.1%) than higher-carbohydrate diets.[14] There is no evidence of significant benefit of these diets on HbA1c or weight change at 12 or 24 months,[14] although the evidence is not precise enough to exclude worthwhile effects on these variables. This apparent diminution in the benefits may be due in part to a decline in dietary adherence over time; additionally, heterogeneity in reporting of medication changes

and adjustment for reduction of these in meta-analyses has been suggested to contribute to an underestimation of effect on glycaemic control.[17] The available evidence has key limitations, including in the variation in definition of what constitutes a 'low-carbohydrate' diet, and there remains a lack of pragmatic trials to demonstrate the feasibility of supporting such diets in routine practice. In most trials to date, the programme has been delivered by specialist staff, offering in-depth advice and support which is unrealistic if this is to be delivered at scale.

In spite of the uncertainties in the evidence base, there is considerable interest from patients and the media in using low-carbohydrate diets in the management of type 2 diabetes[23 24] and some practitioners have begun to recommend patients with type 2 diabetes to follow a low-carbohydrate diet, with or without specific energy restriction.[25] In 2017, the James Lind Alliance identified the role of carbohydrates, dietary change and how best to support people to achieve these changes, as 3 of the top 10 research priorities in type 2 diabetes.[26]

To address this evidence gap, we will conduct an RCT to investigate the feasibility of delivering a low-carbohydrate, low-energy, food-based dietary intervention in primary care, to patients with established type 2 diabetes and body mass index (BMI) ≥30 kg/m$^2$.

The hypothesis is that compared with usual care, an intervention involving targeted health professional advice, goal setting and structured materials, will help patients understand and adhere to a low-carbohydrate, low-energy diet that will lead to significant reductions in weight and associated improvements in glycaemic control (or reduction in diabetic medication required to maintain glycaemic control), providing positive reinforcement of the dietary modification. Few people can maintain a low-energy diet indefinitely (the mean duration of low calorie liquid diet product in the DiRECT study was 16 weeks[9]), but a low-carbohydrate component offers the potential for sustained glycaemic control. Together with the possibility of returning to a low-energy regimen in the case of weight regain, this dietary strategy could provide the foundation for a sustainable approach to the management of this chronic condition.

The specific aims of the Dietary Approaches to the Management Of type 2 Diabetes (DIAMOND) study are to investigate the feasibility of delivering this behavioural and dietary intervention to a population of patients with diabetes in primary care, and determine whether progression to a full-scale RCT is indicated; to assess achievements against a number of process measures to inform future trial design; and to investigate the potential physical, biochemical and economic impact of this behavioural and dietary intervention.

## METHODS AND ANALYSIS
### Design and setting
This feasibility study will be an individually randomised controlled trial, performed in adult patients with type 2

diabetes and a BMI $\geq 30\,kg/m^2$, recruited from general practices in England. Each participant will be enrolled for 3 months from randomisation to final follow-up, and will attend up to seven visits. Due to the nature of the intervention, it will not be possible to blind the participants, clinicians delivering the intervention or some of the study team to the treatment allocation.

## Recruitment

Participants will be recruited from between two and four GP practices for this feasibility study, identified through the regional clinical research network and local expressions of interest. The practices will be asked to search their computerised records to identify people who meet the inclusion criteria. Prior to invitation letters being sent, a general practitioner will screen the list of potential participants to ensure that all those identified are medically appropriate to invite to participate in the trial. The practice will then send potential participants an invitation letter asking them to contact the research team if they are interested, and a participant information sheet with further details about the study.

Participants may also be identified opportunistically during routine consultations or resulting from tests performed as part of the NHS Health Checks programme or diabetes annual review appointments. GPs will provide individuals with the invitation letter and information sheet and invite the individual to ring the study team if they are interested in participating.

Interested individuals will respond to the invitation letter and contact the research team by text, email or telephone. During initial telephone contact, a researcher will discuss the study with the individual and assess self-reported eligibility to participate according to the full inclusion and exclusion criteria as detailed below. They will also establish verbal informed consent for the study team to access the patient's Oxfordshire diabetic eye screening record, to ensure they meet the retinopathy screening inclusion criteria (as this may not always be coded on the GP's electronic notes). Eligible individuals who wish to participate will be booked in for a face-to-face baseline appointment.

## Inclusion criteria
► Participant is willing and able to give informed consent for participation in the study.
► Male or female, aged 18 years or above.
► BMI of $\geq 30\,kg/m^2$.
► Diagnosed with type 2 diabetes, (as defined by an HbA1c $\geq 48\,mmol/mol$ (6.5%) at time of diagnosis).
► Patients must have undergone diabetic retinopathy screening within the last 12 months.

## Exclusion criteria
► History of, or features indicative of, an eating disorder.
► Pregnant, breast feeding, currently undergoing fertility treatment or planning to become pregnant during the course of the study.

► Recent myocardial infarction (MI) or cerebrovascular accident (CVA) (<3 months).
► Uncontrolled ischaemic heart disease, critical ischaemia, uncontrolled hypertension, uncontrolled cardiac arrhythmia, cardiac conduction abnormality (eg, long QT syndrome).
► Cardiac failure (grade II, III or IV New York Heart Association).
► Renal failure (chronic kidney disease (CKD) stage 4 or 5).
► Active treatment for cancer (other than skin cancer treated with curative intent by local treatment only).
► Intercurrent serious infection at time of recruitment.
► Diagnosed with a significant psychiatric disorder or substance abuse.
► Serious neurological disorder, including epilepsy.
► Recently undergone significant surgery (<6 months).
► History of bariatric surgery, including gastric banding.
► Are currently using a 'fasting'/low-energy diet.
► Unwilling to consider any dietary changes.
► Unable to understand English.
► Are currently using insulin or sodium-glucose co-transporter 2 (SGLT2) inhibitor therapy.
► Non-proliferative retinopathy level R2 or more severe (ie, any level more severe than 'background' non-proliferative diabetic retinopathy, R1), proliferative diabetic retinopathy or maculopathy.
► HbA1c $\geq 93\,mmol/mol$ (10.5%).
► Recruiting physician feels they are inappropriate for recruitment due to any other reason.

## Participant flow
Baseline appointments will be conducted with either the practice nurse or a member of the research team at participants' own GP practice, during which informed consent will be sought, eligibility formally assessed and baseline measurements collected. The participants will then be randomised to one of the two trial arms. Participants allocated to the control arm will continue to an appointment with the practice nurse for 'usual care' dietary advice. Participants randomised to receive the intervention will proceed to an appointment with healthcare professionals (practice nurse and then GP) at their practice, during which the intervention materials and advice will be delivered, and medication regimens reviewed for those participants on antihypertensive, hypoglycaemic or lipid-modifying medications. Those in the intervention arm will subsequently be invited to attend for week 2, 4 and 8 study visits, at which further samples and measurements will be obtained. All participants will be invited for a final 12-week follow-up visit, at which time all outcome measures will be repeated. After completion of the 12-week follow-up period, participants will be invited to participate in qualitative focus groups to further explore their experience of this intervention. Participant flow through the study is outlined in figure 1.

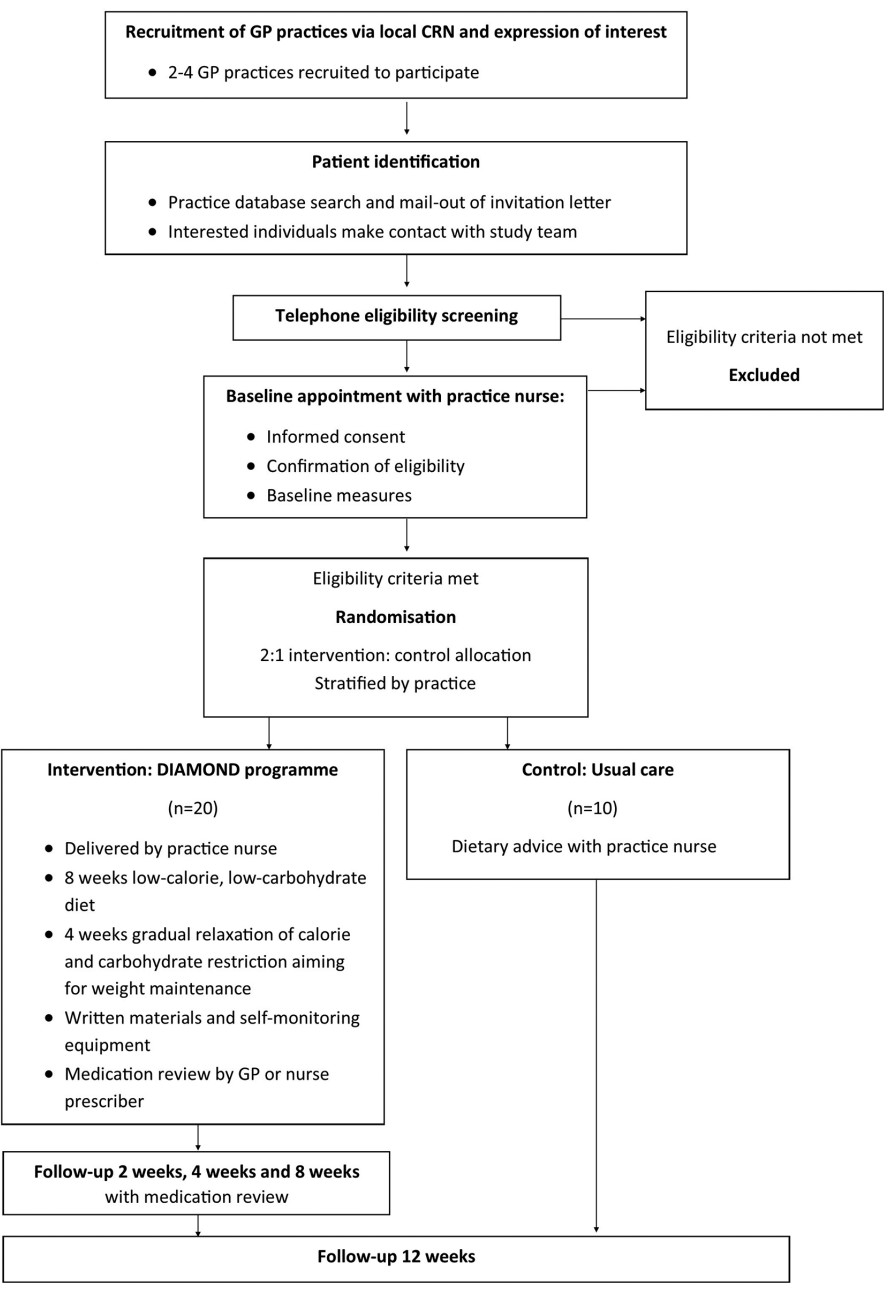

**Figure 1** Participant flow through the study. CRN, clinical research network DIAMOND, Dietary Approaches to the Management Of type 2 Diabetes.

## Sample size

The total number of participants recruited for this study will be 30. As this is a feasibility study, it has not been powered to detect a statistically significant difference in efficacy between the arms. In determining thresholds for progression criteria, based on previous trials of a similar nature, we expect at least 75% achievement.[27 28] If so, the 95% CI will exclude 60%, and therefore we have set this as the progression criteria for each criterion, requiring a sample size of 30 participants allocated in a 2:1 ratio, intervention:control.

## Randomisation

All eligible, consenting participants will be randomised to one of the two trial arms (intervention or control), using permuted block design randomisation with blocks of 3 and 6. Allocation will be stratified by practice. An independent researcher will generate the set of sequences and assign participants to the intervention groups using sequentially numbered sealed envelopes to ensure allocation concealment until interventions are assigned. Due to the nature of this study, it will not be possible to further blind participants, clinicians or some of the study team to the treatment allocation beyond this point.

## Intervention
### The DIAMOND programme

The active intervention is a behaviourally informed, low-carbohydrate, low-energy diet delivered by healthcare professionals at a GP practice. It draws on the motivational value

of the relationship between the healthcare professional and the patient, but provides almost all the technical knowledge-transfer through the use of structured materials such as meal plans, thus addressing the uncertainties and lack of confidence expressed by health professionals about giving dietary advice.[29] The intervention aims to improve patients' adherence to the programme by providing a structured simple behavioural support programme, including goal-setting, planning, feedback and problem-solving. It is designed to induce rapid changes in dietary composition and weight loss, followed by a sustainable maintenance programme. The intervention is envisaged to take 10–15 min of support from the GP and 80 min from the nurse to deliver over four sessions of support (baseline, week 2, week 4 and week 8).

This intervention was designed following the principles of the person-based approach.[30] It involved examining recent systematic reviews, reviewing qualitative studies and process assessments of these types of interventions and developing the behavioural and dietary intervention programme with a multi-disciplinary team comprising a diabetes-specialist dietitian and clinician, primary care physicians and patient involvement. Two groups of patients with type 2 diabetes, including those who had recently tried variations of a low-carbohydrate, low-energy diet, also met to discuss the intervention and to inform and advise on the dietary and behavioural support components.

The dietary component of this intervention consists of a low-carbohydrate and energy restricted diet (800–1000 kcal/day) for 8 weeks, transitioning to a 4-week maintenance period. The core principles include: advice to exclude sugary and starchy foods high in carbohydrates (eg, biscuits, confectionery, bread, pasta, potatoes) entirely from their diet (with the exception of dairy and limited fruit intake), strict portion control and avoiding energy-dense foods. Standardised 'healthy eating' advice regarding fresh vegetables, and lean meat and fish, is also included. After the initial 8-week intensive weight loss phase, participants will be advised to gradually increase their energy intake, reintroducing one 'normal-sized' meal at a time, until they remain weight stable, but with guidance on maintaining a sustainable lower-carbohydrate diet in the longer term.

The design of this intervention has been additionally informed by behavioural analysis, identifying domains of the Behaviour Change Wheel[31] and Theoretical Domains Framework[32] to promote successful behaviour change. Using these frameworks, the key influencing factors which have been targeted are: psychological capability (including knowledge, skills, decision processes and behavioural regulation), social and physical opportunity (social influences and environmental context), reflective motivation (beliefs about capabilities and consequences, optimism, intentions, goals and reinforcement).

To address these components, the intervention will include:

► A brief behavioural intervention consisting of both dietary and motivational advice, including the rationale behind the intervention, to be delivered by a healthcare professional in primary care. The dietary advice includes the essential features of the diet—low energy and low carbohydrate content, as described above.

► Written resources, including advice for meal planning and food selection, and suggested recipes for 'real food' choices—to empower patients with the knowledge and skills for decision-making about food.

► Personalised realistic goal setting (eg, change in weight, HbA1c, number of diabetic medications).

► Diary to record self-assessment of programme adherence.

► Structured healthcare professional follow-up, to provide support and contingency planning, and feedback on progress. Additionally, patients will be able to self-monitor their health with the use of home finger-prick-testing blood glucose meters, and blood pressure (BP) machines, for additional motivation and monitoring of progress.

► Feedback from healthcare professional on personal changes in health parameters as a consequence of the intervention (changes in weight, BP, HbA1c).

## Comparator

Participants randomised to the control arm will receive usual care at their week 0 study visit, comprising a single face-to-face appointment with a healthcare professional, during which they will receive standard dietary and lifestyle information based on the Diabetes UK 'what is a healthy balanced diet for diabetes' leaflet. If their managing GP or nurse feels they warrant referral for further diabetic care or input (eg, diabetes education course), or intensification of medications, and would ordinarily have pursued this as part of their routine care outside of the trial, this will be permitted as forming part of 'usual care', but will be documented.

## Outcomes
### Primary

The primary objective of this study is to test the feasibility of a low-carbohydrate, low-energy, behavioural intervention, comprising targeted advice from a health professional combined with written dietary information, to promote weight loss and improved glycaemic control in patients with type 2 diabetes. This feasibility study will then determine whether to progress to a full randomised control trial.

The following progression criteria will determine whether to progress to a full trial:

1. That 60% of allocated 'intervention' group participants attempt the dietary intervention after randomisation. (Evaluated by documentation after intervention visit; assessed as the proportion of patients who agree to start the dietary intervention, as recorded by the nurse delivering the intervention session.)

2. Fidelity of intervention delivery: That healthcare professionals conduct the intervention delivery session with at least 60% of essential elements present.

(Evaluated by assessment of audio recorded consultations against a checklist of pre-specified essential elements.)

3. That 60% of participants attend the final follow-up session.
   (Documentation of final study visit.)

## Secondary

A mixture of quantitative and qualitative methods will be used to assess process measures and effectiveness measures. This feasibility study is not powered to detect statistical differences in efficacy outcomes but will examine the following parameters in order to test trial procedures, process, resources and management, to aid sample size estimates for a future trial, to determine the most appropriate primary outcome measures for a future trial and to inform further development of the intervention strategy.

## Process measures

► Percentage of eligible patients, as a proportion of the total population of patients in a practice, with type 2 diabetes.
► Percentage of people who fulfil the recruitment criteria who accept the invitation to participate
► Proportion of patients who enrol in the study who are deemed to have with 'suboptimal control' (HbA1c above the National Institute for Health and Care Excellence target of ≥7%).
► Participant adherence to the protocol: including, change in dietary composition (low-carbohydrate, energy restricted—assessed using 24 hours dietary recall questionnaires); participants' self-reported concordance with the intervention; availability of data for outcome measures; attendance at follow-up sessions; contamination of the control group (ie, those who choose to follow the principles of the intervention (ie, follow a low-carbohydrate diet), despite being allocated to the control group). This will be assessed using 24 hours dietary recall questionnaires to establish change in dietary composition (frequency of carbohydrate consumption, energy restriction) of control group participants.
► Difference between 'baseline' HbA1c value and that used from the latest record as inclusion criteria.
► Serious adverse events (SAEs) reported up to the end of the 12-week study participation period.

## Exploratory outcomes

► Change in HbA1c—number of patients previously in diabetic HbA1c range, now in 'at risk of diabetes' (6%–6.4%) or 'diabetes in remission' (<6.0%, off medications) HbA1c range.
► Change in fasting glucose, fasting insulin, Homeostasis Model Assessment steady state beta cell function (HOMA-%B) and insulin sensitivity (HOMA-%S).
► Change in weight.

► Change in diabetic medication (number of diabetic medications currently prescribed to the patient; dose of diabetic medications; initiation of new medication during study period; initiation of insulin; initiation of injectable diabetic medication; number of medications stopped or changed during the study period).
► Change in lipid profile—total cholesterol, high-density lipoprotein (HDL), triglycerides, calculated non-HDL cholesterol and total cholesterol:HDL ratio.
► Change in liver function tests (bilirubin, alanine aminotransferase (ALT), aspartate aminotransferase (AST), alkaline phosphatase (ALP), albumin, AST:ALT).
► Change in BP (systolic, diastolic).
► Change in antihypertensive medication (number of medications started and stopped during study period; dose of antihypertensive medications; initiation of new medication during study period).
► Change in medication prescribing costs (total and diabetic) across study group and total practice diabetic population.
► Effect on patient's Problem Areas in Diabetes (PAID) score.

## Qualitative measures

► Experience of and acceptability of the intervention, for patients and healthcare practitioners (assessed using qualitative focus groups following completion of the initial study period).

## Measurements

Figure 2 provides a summary of the measurements collected.

### Sociodemographic characteristics

Participants will be asked to self-report age, sex and ethnicity, education history and employment status.

### Medical and medication history

Relevant medical history and all current medication will be recorded and checked against the participant's medical record.

### Physical measurements

Height will be measured using stadiometers, to the nearest 1 cm. Weight will be measured to the nearest 0.1 kg using a digital scale (SC-240 MA, Tanita Japan). BMI will be calculated using the standard formula BMI=weight(kg)/height(m).[2] BP will be measured in triplicate after 5 min seated rest, with at least 1 min between each measurement. All assessors will be trained in standardised methods of taking physical measurements according to the study manual of procedures.

### Fasting blood sample

A fasting venous blood sample for HbA1c, fasting insulin, fasting glucose, liver function tests (bilirubin, ALT, AST, ALP, albumin) and lipid profile (total cholesterol, triglycerides, HDL, calculated low-density lipoprotein

| | STUDY PERIOD | | | | | |
|---|---|---|---|---|---|---|
| | Enrolment | Allocation | Study Visits | | | Follow up |
| **TIMEPOINT** | *Baseline* | *0* | *Week 2* | *Week 4* | *Week 8* | *Week 12* |
| **ENROLMENT:** | | | | | | |
| *Eligibility screen* | X | | | | | |
| *Informed consent* | X | | | | | |
| *Randomisation* | X | | | | | |
| **INTERVENTION GROUPS:** | | | | | | |
| *DIAMOND programme* | | X | X | X | X | X |
| *Usual Care* | | X | X | | | X |
| **ASSESSMENTS:** | | | | | | |
| *Demographics* | X | | | | | |
| *Medical history* | X | | | | | X |
| *Medication review* | X | | X | X | X | X |
| *Weight* | X | | X | X | X | X |
| *Height* | X | | X | X | X | X |
| *Blood pressure* | X | | X | X | X | X |
| *Fasting blood sample* | X | | | | X | X |
| *Adverse event assessment* | | | X | X | X | X |
| *24 hour dietary recall questionnaire* | X | | X | | X | X |
| *Motivation assessment* | X | | | | | X |
| *Problem Areas in Diabetes (PAID) score* | X | | | | | X |
| *Self-reported adherence to intervention* | X | | X | X | X | X |

**Figure 2** Schedule of measurements. DIAMOND, Dietary Approaches to the Management Of type 2 Diabetes.

(LDL) and total:HDL cholesterol ratio) will be collected at baseline, 8 weeks and 12 weeks. All blood samples will be taken, handled, analysed and disposed of according to standard NHS procedures and local practice policy.

## Questionnaires

During study follow-up visits, patients will be asked to complete questionnaires to assess the following measures:

▶ *Quality of life:* the PAID score, a 20-item questionnaire measuring problems related to emotions, treatment, food and social support,[33] will be measured at baseline and 12 weeks.

▶ *Dietary composition:* a 24-hour dietary recall questionnaire will be used to assess intake of different food groups at baseline, 2, 8 and 12 weeks.

▶ *Motivation:* self-reported motivation and perceptions across domains of diet, health and diabetes control will be assessed using a 6-point questionnaire at baseline and 12 weeks.

▶ *Self-reported adherence to intervention:* self-reported adherence to the three core components of the dietary intervention will be assessed at 2, 4, 8 and 12 weeks.

## Retention and withdrawal

Each participant will have the right to withdraw from the study at any time. In addition, the investigator may discontinue a participant from the study at any time if they consider it necessary for any reason, including ineligibility (either arising during the study or having been overlooked at screening). If a participant requests to withdraw from the study, it will be explained to them that we would like to use their data collected up to the point at which they have withdrawn from the study, unless they request that we do not do so. The reason for withdrawal will be recorded in the case report form (CRF). Withdrawn participants will not be replaced in this feasibility study. To reflect the burden of participation and promote participant retention and complete follow-up,

participants will be offered a £20 gift card on attending the 12-week follow-up appointment.

## Adverse events

We will record and report all SAEs following Good Clinical Practice and standard Health Research Authority (HRA) processes. The duration of the SAE recording period for each participant lasts from their enrolment on to the study, to their completion of the study. We elected not to record other adverse events (AEs) because this process is burdensome for clinicians and participants, and we considered that this programme was unlikely to induce significant AEs.

## Data management

Data will be recorded on hard copy CRFs and subsequently entered into a web-based data-capture system (RedCap, 2018 Vanderbilt University) with data stored on a secure server hosted by the Primary Care Clinical Trials Unit at the University of Oxford. The system has an inbuilt audit trail facility and ability to run internal validation checks.

## Statistical analysis

Primary outcome measures for this study are progression criteria, to inform any future RCT powered to detect an intervention effect. Analysis for progression criteria will use data from all participants recruited to the trial. Descriptive statistical methods along with inferential statistics presenting CIs will be used to analyse and report progression criteria.

Secondary outcome measures for this study include process measures and exploratory effectiveness measures. Both measures will use all collected data, including that from participants who have not completed the trial, and from participating practices (where pertaining to proportions of patients identified as eligible, and responding to invitations to participate). Descriptive comparative summary statics (eg, difference in means or proportions, together with 95% CIs, comparison with baseline and overall change compared with control group) will be used.

Acceptability and experience of the intervention by participants and healthcare professionals will be collected as qualitative data from interviews and focus groups conducted after intervention completion. These data will be qualitatively analysed using thematic analysis, and reported descriptively.

## Patient involvement

This study evolved through patient demand in the local area, to know more about dietary options and in particular low-carbohydrate diets for people with type 2 diabetes. We convened two panels of patients with type 2 diabetes, one consisting of members who had recently tried variations of a low-carbohydrate, low-energy diet, to inform and advise on the dietary and behavioural support components of the intervention, the patient materials and the perceived benefits or burdens of the study for patients. One patient member has subsequently joined the trial management group and will be involved in trial steering decisions during the monitoring and running of the trial, and regarding progressing to a full-scale trial.

## Qualitative substudy

The purpose of the nested qualitative sub-study is to explore patients' and healthcare professionals' experience of receiving, enacting and delivering this intervention, and of participating in the study. We will seek the views of up to 10 patient participants from the intervention group, from those who consent to be contacted for the nested qualitative study, and 6–8 healthcare professionals. The patient participants chosen will aim to reflect a range of success with weight loss and glycaemic control, with thought given to not mixing those who have found the programme easy or successful, with those who have not achieved their goals. Additionally, we will aim to capture a breadth of participant characteristics (age, gender, social class, ethnicity or, for the healthcare professional interviews, the participant's profession). All healthcare professionals who have been involved in the study will be invited to take part. Should more participants volunteer than are required, purposive sampling will be used to ensure both GPs and nurses are represented in the cohort, and to include those who have conducted a greater number of sessions.

Focus groups will be conducted after all participants have completed their 3-month study follow-up period. Written informed consent will be taken before the focus group commences. All focus groups will be audio recorded, and these audio-recordings will be transcribed. Patients will be reimbursed for their time and practices will be reimbursed for the healthcare professionals taking time out from their clinical duties to participate.

Focus groups will follow broad topic areas based on the study objectives, but will encourage participants to discuss their perceptions and experiences freely and in depth. Topics will include: the acceptability of the intervention; views on which components of their treatment group they felt were effective and which were not effective; thoughts about impact on experience of diabetes and its management; thoughts about weight management going forwards. Data will be analysed using thematic analysis.

## ETHICS AND DISSEMINATION

Any protocol modifications will be submitted for review by the research ethics committee and amended at the trial registry.

If the trial proves feasible, according to the specified progression criteria it is planned that the results will inform design of a full-scale randomised trial to test the efficacy of the intervention to improve glycaemic control.

The findings of this feasibility study will be submitted for publication in a peer-reviewed journal, and presented at conferences, to disseminate the results to academic and health professional audiences, and made available to

participants and to a wider public on our website at the time of publication.

**Acknowledgements** The authors would like to thank the members of the DIAMOND patient participation panels for their invaluable input and advice throughout all stages of study design.

**Contributors** EM, PA and SAJ developed the concept for the study and wrote the first draft of the protocol. EM, PA, SAJ and PD prepared the study documents and coordinated the HRA and ethics application. EM drafted the manuscript for publication, with input from PA and SAJ. PD, MN, CB, RF, KH and GT were involved in the detailed design of the study, and approved the final protocol and manuscript.

**Funding** PA and SAJ are NIHR Senior Investigators and funded by NIHR CLAHRC Oxford. PA, SAJ, GDT, PD and MN are supported by the NIHR Oxford BRC. This work was supported by the National Institute for Health Research (NIHR) Oxford Biomedical Research Centre (BRC) and the School for Primary Care Research (SPCR) (grant reference number 404).

**Disclaimer** This report is independent research by the National Institute for Health Research. The views expressed in this publication are those of the author(s) and not necessarily those of the NHS, the University of Oxford, the NIHR or the Department of Health.

**Competing interests** PA and SAJ have received research grant funding but no personal renumeration from commercial weight loss companies, but none of these companies have interests in the programme described here. CB is the author of "The 8-week blood sugar diet recipe book", published in 2016. PAD is a member of the PHE/SACNE/DUK committee reviewing evidence on the effects of low carbohydrate diets in people with type 2 diabetes.

**Patient consent** Not required.

**Ethics approval** The study protocol (V2.0 1.3.18) was reviewed and approved by the South Central Oxford B REC Committee (Ref: 18/SC/0071).

**Provenance and peer review** Not commissioned; externally peer reviewed.

**Author note** This trial is sponsored by the University of Oxford, Clinical Trials and Research Governance, Joint Research Office, Block 60, Churchill Hospital, Old Road, Headington, Oxford, OX3 7LE, UK.

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
