## [Reviewer comments · BMJ Open]

ARTICLE DETAILS

TITLE (PROVISIONAL)	Dietary approaches to the management of type 2 diabetes (DIAMOND): protocol for a randomised feasibility trial
AUTHORS	Morris, Elizabeth; Aveyard, Paul; Dyson, Pamela; Noreik, Michaela; Bailey, Clare; Fox, Robin; Hoffman, Kathy; Tan, Garry; Jebb, Susan

VERSION 1 – REVIEW

REVIEWER	Prof MEJ Lean University of Glasgow
REVIEW RETURNED	01-Oct-2018

GENERAL COMMENTS	This paper presents the protocol for a pilot study, to examine the feasibility of delivering a low carbohydrate diet in primary care. If a feasibility study is conducted, it will inform the design of a definitive trial. It is useful to publish the protocol for a definitive trial, but I see no value in publishing the protocol for pilot studies. There is no Discussion section in this paper, which might explain the rationale and process behind a protocol development. This would be of value for the protocol of a large definitive trial, but not a pilot study. The Introduction section (line 100-105) describes the results of the DiRECT trial, with 46% of participants achieving remission of type 2 diabetes at 1 year. It goes on to make an extraordinary unsubstantiated, unreferenced, and incorrect statement that the 'total diet replacement approach is not acceptable to some patients, some of whom may prefer a food-based diet'. In the DiRECT trial, not one of the 149 intervention group participants found the total diet replacement approach unacceptable. In the feasibility study for DiRECT, trialing the Counterweight-Plus Programme, some patients did elect to use a food-based diet, but 100% of those patients switched to the formula total diet replacement, using the prepared formula diet, which they found more acceptable. If a trial of low carbohydrate diet is planned, there seems little point in comparing it with a higher carbohydrate diet in RCT design. There are many head-to-head comparisons which show zero or trivial differences, both in diabetic and non-diabetic subjects. What would be really helpful would be an n=1 randomised trial approach, which allows patients to try both approaches and find which one is best for them. The DIETFITS study illustrated rather elegantly how, within a study population, there are people who do incredibly well with low-carbohydrate or with low-fat diets, while others do poorly, .
--

	If such a study is conducted, It will be important to use the best-evidenced approach as a comparator. That is management with a formula total diet replacement. People with type 2 diabetes have been offered inadequate dietary advice for generations. We now do a have an approach using formula total diet replacement followed by a well researched structured programme to minimise regain, delivered in primary care, which offers a major advantage for patients. That will not be the last word, particularly for long-term weight loss maintenance, and future research has the capacity to improve in many ways. Flexibility over individual diet composition (including reducing glycaemic load, and varying fat/carbohydrate content) may be advantageous, but the current DiRECT intervention itself is not 'unsustainable'. It may even cost less to healthcare services than the present drug-based management, followed by vascular complications and ultimate disability.
--	---

REVIEWER	Roy Taylor Newcastle university, UK
REVIEW RETURNED	29-Oct-2018

GENERAL COMMENTS	The manuscript provides a clear description of a pilot study protocol. It addresses an important question. I have few comments:  1. The statement in lines 141-143 (Few people can 142 sustain a low energy diet beyond 8-12 weeks but the low-carbohydrate component offers the 143 potential for sustained glycaemic control) is incorrect. This common assumption was disproven most clearly in DiRECT in which the fixed 12 week low calorie period was followed by an optional period of either return to normal eating or continuation if wished (to a max of 20 weeks) by the individual for their personal goals. The majority of the 149 Intervention group participants extended such that the mean time on the low calorie liquid diet product was 16 weeks. The statement should be readjusted to reflect this solid observation. 2. It is surprising that weight loss is not a primary outcome of the pilot. If weight loss is inadequate, surely there would be no point in going on to a full study? Certainly the process outcomes are appropriate for this pilot, but the stop/go nature of the hard end point desired must surely be recognised?
---

VERSION 1 – AUTHOR RESPONSE

Reviewer: 1.

Reviewer Name: Prof Mike Lean

Institution and Country: University of Glasgow

Please state any competing interests or state 'None declared': None declared

"This paper presents the protocol for a pilot study, to examine the feasibility of delivering a low carbohydrate diet in primary care. If a feasibility study is conducted, it will inform the design of a definitive trial. It is useful to publish the protocol for a definitive trial, but I see no value in publishing the protocol for pilot studies."

Thank you for your comment. We noted the BMJ Open editorial policy encouraging publication of study protocols, to make available more information than is currently required by trial registries, and

their recent decisions in 2018 to include a wide range of pilot study protocols across the fields of health education and patient care(1-3), and we think that the DIAMOND study protocol will be of relevance and interest to their readership. In addition, we feel that the processes we followed for intervention development with core involvement of PPI panels and multidisciplinary teams might be of interest to the readership, and we were able to describe this in greater detail than in the prospective trial registry.

"There is no Discussion section in this paper, which might explain the rationale and process behind a protocol development. This would be of value for the protocol of a large definitive trial, but not a pilot study."

We have explained the rationale for the trial in the introduction and have followed the BMJ Open advice for authors for the recommended sections to include in a study protocol manuscript.

"The Introduction section (line 100-105 describes the results of the DiRECT trial, with 46% of participants achieving remission of type 2 diabetes at 1 year. It goes on to make an extraordinary unsubstantiated, unreferenced, and incorrect statement that the 'total diet replacement approach is not acceptable to some patients, some of whom may prefer a food-based diet'. In the DiRECT trial, not one of the 149 intervention group participants found the total diet replacement approach unacceptable. In the feasibility study for DiRECT, trialing the Counterweight-Plus Programme, some patients did elect to use a food-based diet, but 100% of those patients switched to the formula total diet replacement, using the prepared formula diet, which they found more acceptable."

We agree that DiRECT shows clearly that this approach is a valuable therapeutic option for people with diabetes, however, as the reviewer discusses below, it is unlikely that a "one size fits all" approach would work for all individuals with type 2 diabetes and overweight or obesity. We acknowledge that acceptability was high among participants allocated to a TDR programme in DiRECT and in our own DROPLET trial(4), but many people approached to participate in both trials did not do so. For example in DiRECT, 1204 patients were invited but not enrolled (that is, 80% of those initially screened for eligibility; of whom 246 declined to take part, 841 did not respond, and 117 were excluded at screening). As clinicians, we also need to consider whether other approaches may also be acceptable and effective. In the PPI work we undertook when designing this study, and the DIAMOND programme intervention development, we sought the views of a diverse range of people living with type 2 diabetes, of different ages, genders and ethnicities. While they were universally enthused by the concept of remission from type 2 diabetes now being achievable, thanks to the findings and publicity of the DiRECT study, and were committed to re-shaping education on this subject both for patients with diabetes, and their physicians, many of them expressed a view that they would not consider undertaking a TDR approach – even if remission from diabetes were the goal – and they wanted to know if this would be achievable with "real food". This patient involvement was central to our development of a protocol to explore whether a dietary approach using a food-based low-calorie diet would be feasible in routine care.

"If a trial of low carbohydrate diet is planned, there seems little point in comparing it with a higher carbohydrate diet in RCT design. There are many head-to-head comparisons which show zero or trivial differences, both in diabetic and non-diabetic subjects. What would be really helpful would be an n=1 randomised trial approach, which allows patients to try both approaches and find which one is best for them. The DIETFITS study illustrated rather elegantly how, within a study population, there are people who do incredibly well with low-carbohydrate or with low-fat diets, while others do poorly. . If such a study is conducted, it will be important to use the best-evidenced approach as a comparator. That is management with a formula total diet replacement. People with type 2 diabetes have been offered inadequate dietary advice for generations. We now do have an approach using formula total diet replacement followed by a well researched structured programme to minimise regain, delivered in

primary care, which offers a major advantage for patients. That will not be the last word, particularly for long-term weight loss maintenance, and future research has the capacity to improve in many ways. Flexibility over individual diet composition (including reducing glycaemic load, and varying fat/carbohydrate content) may be advantageous, but the current DiRECT intervention itself is not 'unsustainable'. It may even cost less to healthcare services than the present drug-based management, followed by vascular complications and ultimate disability. "

We do believe that a trial that compares usual care versus a low-energy, low-carbohydrate diet that we have proposed here is warranted. Firstly, NICE guidance is to eat a healthy diet that is neither particularly low or high in carbohydrate or fat and this is the natural comparator in any pragmatic trial such as this. Secondly, Diabetes UK called for more evidence on the effectiveness of research aiming to induce remission with low energy 'intensive' weight loss programmes, which is what our proposal describes, and low-carbohydrate diets. We believe that our trial will make a useful impact because we have taken a good deal of care to co-create an intervention with patients and make supporting materials for the programme. Diet prescriptions are easy to give but often suffer from low adherence and our programme aims to improve adherence through careful design and therefore this may show a difference, notwithstanding other programmes have showed relatively modest differences. We have amended wording with any implication that the TDR approach is unsustainable to reflect the findings of DiRECT (see below).

This is, of course, not the only trial or intervention which could be considered to improve the management of patients with type 2 diabetes and as we discuss above, the overarching goal is to provide high quality trial evidence of the effectiveness of a number of different approaches so that clinicians can have informed discussions with patients about the nature and effectiveness of different strategies in planning treatments for individuals.

Reviewer: 2

Reviewer Name: Roy Taylor

Institution and Country: Newcastle university, UK

Please state any competing interests or state 'None declared': I am co-lead of DiRECT

"The manuscript provides a clear description of a pilot study protocol. It addresses an important question. I have few comments:

The statement in lines 141-143 (Few people can 142 sustain a low energy diet beyond 8-12 weeks but the low-carbohydrate component offers the 143 potential for sustained glycaemic control) is incorrect. This common assumption was disproven most clearly in DiRECT in which the fixed 12 week low calorie period was followed by an optional period of either return to normal eating or continuation if wished (to a max of 20 weeks) by the individual for their personal goals. The majority of the 149 Intervention group participants extended such that the mean time on the low calorie liquid diet product was 16 weeks. The statement should be readjusted to reflect this solid observation. "

Thank you for your comment and helpful clarification. We have amended the statement to reflect this data; it now reads:

141 "Few people can maintain a low energy diet indefinitely, (the mean duration of low-calorie liquid diet product in the DiRECT study was 16 weeks)(5) but a low-carbohydrate component offers the potential for sustained glycaemic control"

"2. It is surprising that weight loss is not a primary outcome of the pilot. If weight loss is inadequate, surely there would be no point in going on to a full study? Certainly the process outcomes are appropriate for this pilot, but the stop/go nature of the hard end point desired must surely be recognised?"

We agree that the weight loss patients achieve in this pilot study is important for diabetes remission and will report it as an outcome. However, we did not have sufficient funding to run a study large enough to have sufficient power to be able to set a stop-go criterion based on weight. We would like to reassure the reviewer that we will not proceed to seek funding for a definitive trial of this programme (should we pass our feasibility criteria) if all of the process and exploratory outcomes (which include weight) look unpromising.

REFERENCES

1. Nicholson J, Wright SM, Carlisle AM. Pre-post, mixed-methods feasibility study of the WorkingWell mobile support tool for individuals with serious mental illness in the USA: a pilot study protocol. *BMJ open*. 2018;8(2).
2. Thompson TP, Callaghan L, Hazeldine E, Quinn C, Walker S, Byng R, et al. Health trainer-led motivational intervention plus usual care for people under community supervision compared with usual care alone: a study protocol for a parallel-group pilot randomised controlled trial (STRENGTHEN). *BMJ open*. 2018;8(6).
3. Signorelli C, Wakefield CE, Johnston KA, Fardell JE, Brierley M-EE, Thornton-Benko E, et al. 'Re-engage' pilot study protocol: a nurse-led eHealth intervention to re-engage, educate and empower childhood cancer survivors. *BMJ open*. 2018;8(4).
4. Astbury NM, Aveyard P, Nickless A, Hood K, Corfield K, Lowe R, et al. Doctor Referral of Overweight People to Low Energy total diet replacement Treatment (DROPLET): pragmatic randomised controlled trial. *BMJ*. 2018;362.
5. Lean MEJ, Leslie WS, Barnes AC, Brosnahan N, Thom G, McCombie L, et al. Primary care-led weight management for remission of type 2 diabetes (DiRECT): an open-label, cluster-randomised trial. *The Lancet*. 2018;391(10120):541-51.